# AIDE: An algorithm for measuring the accuracy of probabilistic inference algorithms

**Marco F. Cusumano-Towner**
Probabilistic Computing Project
Massachusetts Institute of Technology
marcoct@mit.edu

**Vikash K. Mansinghka**
Probabilistic Computing Project
Massachusetts Institute of Technology
vkm@mit.edu

## Abstract

Approximate probabilistic inference algorithms are central to many fields. Examples include sequential Monte Carlo inference in robotics, variational inference in machine learning, and Markov chain Monte Carlo inference in statistics. A key problem faced by practitioners is measuring the accuracy of an approximate inference algorithm on a specific data set. This paper introduces the auxiliary inference divergence estimator (AIDE), an algorithm for measuring the accuracy of approximate inference algorithms. AIDE is based on the observation that inference algorithms can be treated as probabilistic models and the random variables used within the inference algorithm can be viewed as auxiliary variables. This view leads to a new estimator for the symmetric KL divergence between the approximating distributions of two inference algorithms. The paper illustrates application of AIDE to algorithms for inference in regression, hidden Markov, and Dirichlet process mixture models. The experiments show that AIDE captures the qualitative behavior of a broad class of inference algorithms and can detect failure modes of inference algorithms that are missed by standard heuristics.

## 1 Introduction

Approximate probabilistic inference algorithms are central to diverse disciplines, including statistics, robotics, machine learning, and artificial intelligence. Popular approaches to approximate inference include sequential Monte Carlo, variational inference, and Markov chain Monte Carlo. A key problem faced by practitioners is measuring the accuracy of an approximate inference algorithm on a specific data set. The accuracy is influenced by complex interactions between the specific data set in question, the model family, the algorithm tuning parameters such as the number of iterations, and any associated proposal distributions and/or approximating variational family. Unfortunately, practitioners assessing the accuracy of inference have to rely on heuristics that are either brittle or specialized for one type of algorithm [1], or both. For example, log marginal likelihood estimates can be used to assess the accuracy of sequential Monte Carlo and variational inference, but these estimates can fail to significantly penalize an algorithm for missing a posterior mode. Expectations of probe functions do not assess the full approximating distribution, and they require design specific to each model.

This paper introduces an algorithm for estimating the symmetrized KL divergence between the output distributions of a broad class of exact and approximate inference algorithms. The key idea is that inference algorithms can be treated as probabilistic models and the random variables used within the inference algorithm can be viewed as latent variables. We show how sequential Monte Carlo, Markov chain Monte Carlo, rejection sampling, and variational inference can be represented in a common mathematical formalism based on two new concepts: *generative inference models* and *meta-inference algorithms*. Using this framework, we introduce the Auxiliary Inference Divergence Estimator (AIDE), which estimates the symmetrized KL divergence between the output distributions

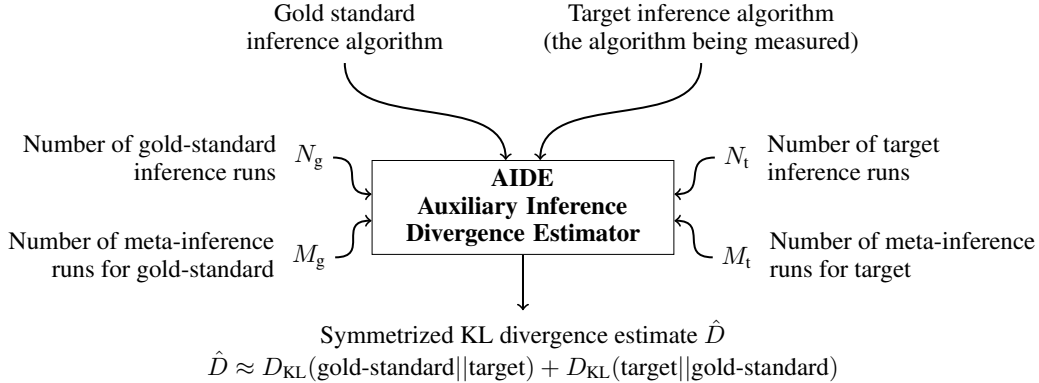

Figure 1: Using AIDE to estimate the accuracy of a target inference algorithm relative to a gold-standard inference algorithm. AIDE is a Monte Carlo estimator of the symmetrized Kullback-Leibler (KL) divergence between the output distributions of two inference algorithms. AIDE uses *meta-inference*: inference over the internal random choices made by an inference algorithm.

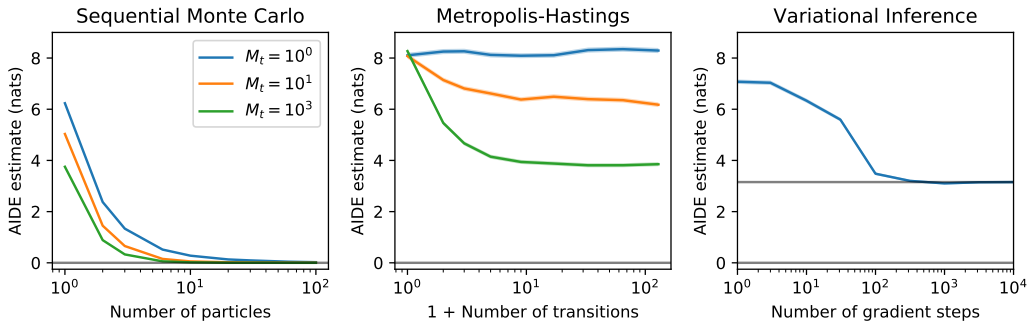

Figure 2: **AIDE applies to SMC, variational, and MCMC algorithms.** Left: AIDE estimates for SMC converge to zero, as expected. Right: AIDE estimates for variational inference converge to a nonzero asymptote that depends on the variational family. Middle: The symmetrized divergence between MH and the posterior converges to zero, but AIDE over-estimates the divergence in expectation. Although increasing the number of meta-inference runs $M_t$ reduces the bias of AIDE, AIDE is not yet practical for measuring MH accuracy due to inaccurate meta-inference for MH.

of two inference algorithms that have both been endowed with a meta-inference algorithm. We also show that the conditional SMC update of Andrieu et al. [2] and the reverse AIS Markov chain of Grosse et al. [3] are both special cases of a 'generalized conditional SMC update', which we use as a canonical meta-inference algorithm for SMC. AIDE is a practical tool for measuring the accuracy of SMC and variational inference algorithms relative to gold-standard inference algorithms. Note that this paper does not provide a practical solution to the MCMC convergence diagnosis problem. Although in principle AIDE can be applied to MCMC, to do so in practice will require more accurate meta-inference algorithms for MCMC to be developed.

## 2 Background

Consider a generative probabilistic model with latent variables $X$ and observed variables $Y$. We denote assignments to these variables by $x \in \mathcal{X}$ and $y \in \mathcal{Y}$. Let $p(x, y)$ denote the joint distribution of the generative model. The posterior distribution is $p(x|y) := p(x, y)/p(y)$ where $p(y) = \sum_x p(x, y)$ is the marginal likelihood, or 'evidence'.

Sampling-based approximate inference strategies including Markov chain Monte Carlo (MCMC, [4, 5]), sequential Monte Carlo (SMC, [6]), annealed importance sampling (AIS, [7]) and importance sampling with resampling (SIR, [8, 9]), generate samples of the latent variables that are approximately distributed according to $p(x|y)$. Use of a sampling-based inference algorithm is often motivated by

theoretical guarantees of exact convergence to the posterior in the limit of infinite computation (e.g. number of transitions in a Markov chain, number of importance samples in SIR). However, how well the sampling distribution approximates the posterior distribution for finite computation is typically difficult to analyze theoretically or estimate empirically with confidence.

Variational inference [10] explicitly minimizes the approximation error of the approximating distribution $q_\theta(x)$ over parameters $\theta$ of a variational family. The error is usually quantified using the Kullback-Leibler (KL) divergence from the approximation $q_\theta(x)$ to the posterior $p(x|y)$, denoted $D_{\mathrm{KL}}(q_\theta(x) \| p(x|y))$. Unlike sampling-based approaches, variational inference does not generally give exact results for infinite computation because the variational family does not include the posterior. Minimizing the KL divergence is performed by maximizing the 'evidence lower bound' (ELBO) $\mathcal{L} = \log p(y) - D_{\mathrm{KL}}(q_\theta(x) \| p(x|y))$ over $\theta$. Since $\log p(y)$ is usually unknown, the actual error (the KL divergence) of a variational approximation is also unknown.

# 3    Estimating the symmetrized KL divergence between inference algorithms

This section defines our mathematical formalism for analyzing inference algorithms; shows how to represent SMC, MCMC, rejection sampling, and variational inference in this formalism; and introduces the Auxiliary Inference Divergence Estimator (AIDE), an algorithm for estimating the symmetrized KL divergence between two inference algorithms.

## 3.1    Generative inference models and meta-inference algorithms

We define an inference algorithm as a procedure that returns a single approximate posterior sample. Repeated runs of the algorithm give independent samples. The algorithm has an 'output distribution' $q(x)$ that gives the probability of returning $x$. Note that the dependence of $q(x)$ on the observations $y$ that define the inference problem is suppressed in the notation. The algorithm is accurate when $q(x) \approx p(x|y)$ for all $x$. We denote a sample returned from the algorithm by $x \sim q(x)$.

A naive simple Monte Carlo estimator of the KL divergence between the output distributions of two inference algorithms requires evaluating output probabilities for both algorithms. However, it is typically intractable to compute output probabilities for sampling-based inference algorithms like MCMC and SMC, because that would require marginalizing over all possible values that the random variables drawn during the algorithm could possibly take. A similar difficulty arises when computing the marginal likelihood $p(y)$ of a generative probabilistic model $p(x, y)$. This suggests that we treat the inference algorithm as a generative model, estimate its output probabilities using ideas from marginal likelihood estimation, and use these estimates in a Monte Carlo estimator of the divergence. We begin by making the analogy between an inference algorithm and a generative model explicit:

**Definition 3.1** (Generative inference model)**.** A *generative inference model* is a tuple $(\mathcal{U}, \mathcal{X}, q)$ where $q(u, x)$ is a joint distribution defined on $\mathcal{U} \times \mathcal{X}$. A generative inference model *models* an inference algorithm if the output probability of the inference algorithm is the marginal likelihood $q(x) = \sum_u q(u, x)$ of the model for all $x$. An element $u \in \mathcal{U}$ represents a complete assignment to the internal random variables within the inference algorithm, and is called a 'trace'. The ability to simulate from $q(u, x)$ is required, but the ability to compute the probability $q(u, x)$ is not. A simulation, denoted $u, x \sim q(u, x)$, may be obtained by running the inference algorithm and recording the resulting trace $u$ and output $x$.[1]

A generative inference model can be understood as a generative probabilistic model where the $u$ are the latent variables and the $x$ are the observations. Note that two different generative inference models may use different representations for the internal random variables of the same inference algorithm. In practice, constructing a generative inference model from an inference algorithm amounts to defining the set of internal random variables. For marginal likelihood estimation in a generative inference model, we use a 'meta-inference' algorithm:

**Definition 3.2** (Meta-inference algorithm)**.** For a given generative inference model $(\mathcal{U}, \mathcal{X}, q)$, a *meta-inference algorithm* is a tuple $(r, \xi)$ where $r(u; x)$ is a distribution on traces $u \in \mathcal{U}$ of the inference algorithm, indexed by outputs $x \in \mathcal{X}$ of the inference algorithm, and where $\xi(u, x)$ is the

following function of $u$ and $x$ for some $Z > 0$:

$$\xi(u,x) := Z \frac{q(u,x)}{r(u;x)} \tag{1}$$

We require the ability to sample $u \sim r(u;x)$ given a value for $x$, and the ability to evaluate $\xi(u,x)$ given $u$ and $x$. We call a procedure for sampling from $r(u;x)$ a 'meta-inference sampler'. We do not require the ability to evaluate the probability $r(u;x)$.

A meta-inference algorithm is considered accurate for a given $x$ if $r(u;x) \approx q(u|x)$ for all $u$. Conceptually, a meta-inference sampler tries to answer the question 'how could my inference algorithm have produced this output $x$?' Note that if it is tractable to evaluate the marginal likelihood $q(x)$ of the generative inference model up to a normalizing constant, then it is not necessary to represent internal random variables for the inference algorithm, and a generative inference model can define the trace as an empty token $u = ()$ with $\mathcal{U} = \{()\}$. In this case, the meta-inference algorithm has $r(u;x) = 1$ for all $x$ and $\xi(u,x) = Zq(x)$.

### 3.2 Examples

We now show how to construct generative inference models and corresponding meta-inference algorithms for SMC, AIS, MCMC, SIR, rejection sampling, and variational inference. The meta-inference algorithms for AIS, MCMC, and SIR are derived as special cases of a generic SMC meta-inference algorithm.

**Sequential Monte Carlo.** We consider a general class of SMC samplers introduced by Del Moral et al. [6], which can be used for approximate inference in both sequential state space and non-sequential models. We briefly summarize a slightly restricted variant of the algorithm here, and refer the reader to the supplement and Del Moral et al. [6] for full details. The SMC algorithm propagates $P$ weighted particles through $T$ steps, using proposal kernels $k_t$ and multinomial resampling based on weight functions $w_1(x_1)$ and $w_t(x_{t-1}, x_t)$ for $t > 1$ that are defined in terms of 'backwards kernels' $\ell_t$ for $t = 2 \ldots T$. Let $x_t^i$, $w_t^i$ and $W_t^i$ denote the value, unnormalized weight, and normalized weight of particle $i$ at time $t$, respectively. We define the output sample $x$ of SMC as a single draw from the particle approximation at the final time step, which is obtained by sampling a particle index $I_T \sim \text{Categorical}(W_T^{1:P})$ where $W_T^{1:P}$ denotes the vector of weights $(W_T^1, \ldots, W_T^P)$, and then setting $x \leftarrow x_T^{I_T}$. The generative inference model uses traces of the form $u = (\mathbf{x}, \mathbf{a}, I_T)$, where $\mathbf{x}$ contains the values of all particles at all time steps and where $\mathbf{a}$ (for 'ancestor') contains the index $a_t^i \in \{1 \ldots P\}$ of the parent of particle $x_{t+1}^i$ for each particle $i$ and each time step $t = 1 \ldots T - 1$. Algorithm 1 defines a canonical meta-inference sampler for this generative inference model that takes as input a latent sample $x$ and generates an SMC trace $u \sim r(u;x)$ as output. The meta-inference sampler first generates an ancestral trajectory of particles $(x_1^{I_1}, x_2^{I_2}, \ldots, x_T^{I_T})$ that terminates in the output sample $x$, by sampling sequentially from the backward kernels $\ell_t$, starting from $x_T^{I_T} = x$. Next, it runs a conditional SMC update [2] conditioned on the ancestral trajectory. For this choice of $r(u;x)$ and for $Z = 1$, the function $\xi(u,x)$ is closely related to the marginal likelihood estimate $\widehat{p(y)}$ produced by the SMC scheme:[2] $\xi(u,x) = p(x,y)/\widehat{p(y)}$. See supplement for derivation.

**Annealed importance sampling.** When a single particle is used ($P = 1$), and when each forward kernel $k_t$ satisfies detailed balance for some intermediate distribution, the SMC algorithm simplifies to annealed importance sampling (AIS, [7]), and the canonical SMC meta-inference inference (Algorithm 1) consists of running the forward kernels in reverse order, as in the reverse annealing algorithm of Grosse et al. [3, 12]. The canonical meta-inference algorithm is accurate ($r(u;x) \approx q(u;x)$) if the AIS Markov chain is kept close to equilibrium at all times. This is achieved if the intermediate distributions form a sufficiently fine-grained sequence. See supplement for analysis.

**Markov chain Monte Carlo.** We define each run of an MCMC algorithm as producing a single output sample $x$ that is the iterate of the Markov chain produced after a predetermined number of burn-in steps has passed. We also assume that each MCMC transition operator satisfies detailed balance

**Algorithm 1** Generalized conditional SMC (a canonical meta-inference sampler for SMC)

---

**Require:** Latent sample $x$, SMC parameters
  $I_T \sim \text{Uniform}(1 \ldots P)$
  $x_T^{I_T} \leftarrow x$
  **for** $t \leftarrow T - 1 \ldots 1$ **do**
    $I_t \sim \text{Uniform}(1 \ldots P)$
    ▷ Sample from backward kernel
    $x_t^{I_t} \sim \ell_{t+1}(\cdot; x_{t+1}^{I_{t+1}})$
  **for** $i \leftarrow 1 \ldots P$ **do**
    **if** $i \neq I_1$ **then** $x_1^i \sim k_1(\cdot)$
    $w_1^i \leftarrow w_1(x_1^i)$
  **for** $t \leftarrow 2 \ldots T$ **do**
    $W_{t-1}^{1:P} \leftarrow w_{t-1}^{1:P} / (\sum_{i=1}^{P} w_{t-1}^i)$
    **for** $i \leftarrow 1 \ldots P$ **do**
      **if** $i = I_t$ **then** $a_{t-1}^i \leftarrow I_{t-1}$
      **else**
        $a_{t-1}^i \sim \text{Categorical}(W_{t-1}^{1:P})$
        $x_t^i \sim k_t(\cdot; x_{t-1}^{a_{t-1}^i})$
      $w_t^i \leftarrow w_t(x_{t-1}^{a_{t-1}^i}, x_t^i)$
  $u \leftarrow (\mathbf{x}, \mathbf{a}, I_T)$   ▷ Return an SMC trace
  **return** $u$

---

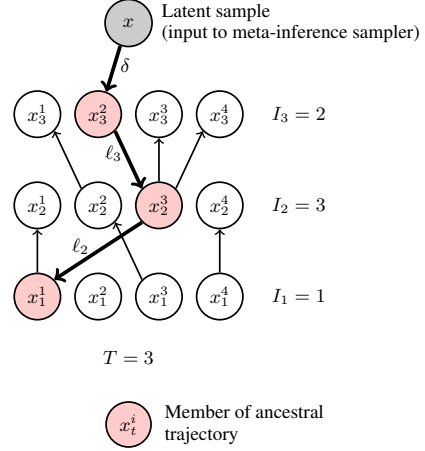

with respect to the posterior $p(x|y)$. Then, this is formally a special case of AIS. However, unless the Markov chain was initialized near the posterior $p(x|y)$, the chain will be far from equilibrium during the burn-in period, and the AIS meta-inference algorithm will be inaccurate.

**Importance sampling with resampling.** Importance sampling with resampling, or SIR [8] can be seen as a special case of SMC if we set the number of steps to one ($T = 1$). The trace of the SIR algorithm is then the set of particles $x_1^i$ for $i \in \{1, \ldots, P\}$ and output particle index $I_1$. Given output sample $x$, the canonical SMC meta-inference sampler then simply samples $I_1 \sim \text{Uniform}(1 \ldots P)$, sets $x_1^{I_1} \leftarrow x$, and samples the other $P - 1$ particles from the importance distribution $k_1(x)$.

**Rejection sampling.** To model a rejection sampler for a posterior distribution $p(x|y)$, we assume it is tractable to evaluate the unnormalized posterior probability $p(x, y)$. We define $\mathcal{U} = \{()\}$ as described in Section 3.1. For meta-inference, we define $Z = p(y)$ so that $\xi(u, x) = p(y)p(x|y) = p(x, y)$. It is not necessary to represent the internal random variables of the rejection sampler.

**Variational inference.** We suppose a variational approximation $q_\theta(x)$ has been computed through optimization over the variational parameters $\theta$. We assume that it is possible to sample from the variational approximation, and evaluate its normalized probability distribution. Then, we use $\mathcal{U} = \{()\}$ and $Z = 1$ and $\xi(u, x) = q_\theta(x)$. This formulation also applies to amortized variational inference algorithms, which reuse the parameters $\theta$ for inference across observation contexts $y$.

### 3.3 The auxiliary inference divergence estimator

Consider a probabilistic model $p(x, y)$, a set of observations $y$, and two inference algorithms that approximate $p(x|y)$. One of the two inference algorithms is considered the 'gold-standard', and has a generative inference model $(\mathcal{U}, \mathcal{X}, q_g)$ and a meta-inference algorithm $(r_g, \xi_g)$. The second algorithm is considered the 'target' algorithm, with a generative inference model $(\mathcal{V}, \mathcal{X}, q_t)$ (we denote a trace of the target algorithm by $v \in \mathcal{V}$), and a meta-inference algorithm $(r_t, \xi_t)$. This section shows how to estimate an upper bound on the symmetrized KL divergence between $q_g(x)$ and $q_t(x)$, which is:

$$D_{\text{KL}}(q_g(x) \parallel q_t(x)) + D_{\text{KL}}(q_t(x) \parallel q_g(x)) = \mathbb{E}_{x \sim q_g(x)} \left[ \log \frac{q_g(x)}{q_t(x)} \right] + \mathbb{E}_{x \sim q_t(x)} \left[ \log \frac{q_t(x)}{q_g(x)} \right] \quad (2)$$

We take a Monte Carlo approach. Simple Monte Carlo applied to the Equation (2) requires that $q_g(x)$ and $q_t(x)$ can be evaluated, which would prevent the estimator from being used when either inference algorithm is sampling-based. Algorithm 2 gives the Auxiliary Inference Divergence Estimator

(AIDE), an estimator of the symmetrized KL divergence that only requires evaluation of $\xi_g(u, x)$ and $\xi_t(v, x)$ and not $q_g(x)$ or $q_t(x)$, permitting its use with sampling-based inference algorithms.

---

**Algorithm 2** Auxiliary Inference Divergence Estimator (AIDE)

---

**Require:** 
Gold-standard inference model and meta-inference algorithm $\quad(\mathcal{U}, \mathcal{X}, q_g)$ and $(r_g, \xi_g)$
Target inference model and meta-inference algorithm $\quad(\mathcal{V}, \mathcal{X}, q_t)$ and $(r_t, \xi_t)$
Number of runs of gold-standard algorithm $\quad N_g$
Number of runs of meta-inference sampler for gold-standard $\quad M_g$
Number of runs of target algorithm $\quad N_t$
Number of runs of meta-inference sampler for target $\quad M_t$

**for** $n \leftarrow 1 \ldots N_g$ **do**
$\quad u_{n,1}, x_n \sim q_g(u, x) \quad \triangleright$ Run gold-standard algorithm, record trace $u_{n,1}$ and output $x_n$
$\quad$**for** $m \leftarrow 2 \ldots M_g$ **do**
$\quad\quad u_{n,m} \sim r_g(u; x_n) \quad \triangleright$ Run meta-inference sampler for gold-standard algorithm, on input $x_n$
$\quad$**for** $m \leftarrow 1 \ldots M_t$ **do**
$\quad\quad v_{n,m} \sim r_t(v; x_n) \quad \triangleright$ Run meta-inference sampler for target algorithm, on input $x_n$
**for** $n \leftarrow 1 \ldots N_t$ **do**
$\quad v'_{n,1}, x'_n \sim q_t(v, x) \quad \triangleright$ Run target algorithm, record trace $v'_{n,1}$ and output $x'_n$
$\quad$**for** $m \leftarrow 2 \ldots M_t$ **do**
$\quad\quad v'_{n,m} \sim r_t(v; x'_n) \quad \triangleright$ Run meta-inference sampler for target algorithm, on input $x'_n$
$\quad$**for** $m \leftarrow 1 \ldots M_g$ **do**
$\quad\quad u'_{n,m} \sim r_g(u; x'_n) \quad \triangleright$ Run meta-inference sampler for gold-standard algorithm, on input $x'_n$
$\hat{D} \leftarrow \dfrac{1}{N_g} \sum_{n=1}^{N_g} \log \left( \dfrac{\frac{1}{M_g} \sum_{m=1}^{M_g} \xi_g(u_{n,m}, x_n)}{\frac{1}{M_t} \sum_{m=1}^{M_t} \xi_t(v_{n,m}, x_n)} \right) + \dfrac{1}{N_t} \sum_{n=1}^{N_t} \log \left( \dfrac{\frac{1}{M_t} \sum_{m=1}^{M_t} \xi_t(v'_{n,m}, x'_n)}{\frac{1}{M_g} \sum_{m=1}^{M_g} \xi_g(u'_{n,m}, x'_n)} \right)$
**return** $\hat{D}$ $\qquad\qquad\qquad\qquad \triangleright \hat{D}$ is an estimate of $D_{\mathrm{KL}}(q_g(x) \| q_t(x)) + D_{\mathrm{KL}}(q_t(x) \| q_g(x))$

---

The generic AIDE algorithm above is defined in terms of abstract generative inference models and meta-inference algorithms. For concreteness, the supplement contains the AIDE algorithm specialized to the case when the gold-standard is AIS and the target is a variational approximation.

**Theorem 1.** *The estimate $\hat{D}$ produced by AIDE is an upper bound on the symmetrized KL divergence in expectation, and the expectation is nonincreasing in AIDE parameters $M_g$ and $M_t$.*

See supplement for proof. Briefly, AIDE estimates an upper bound on the symmetrized divergence in expectation because it uses unbiased estimates of $q_t(x_n)$ and $q_g(x_n)^{-1}$ for $x_n \sim q_g(x)$, and unbiased estimates of $q_g(x'_n)$ and $q_t(x'_n)^{-1}$ for $x'_n \sim q_t(x)$. For $M_g = 1$ and $M_t = 1$, AIDE over-estimates the true symmetrized divergence by:

$$
\mathbb{E}[\hat{D}] - (D_{\mathrm{KL}}(q_g(x) \| q_t(x)) + D_{\mathrm{KL}}(q_t(x) \| q_g(x))) =
$$
$$
\left( \begin{array}{l} \quad \mathbb{E}_{x \sim q_g(x)} \left[ D_{\mathrm{KL}}(q_g(u|x) \| r_g(u; x)) + D_{\mathrm{KL}}(r_t(v; x) \| q_t(v|x)) \right] \\ + \ \mathbb{E}_{x \sim q_t(x)} \left[ D_{\mathrm{KL}}(q_t(v|x) \| r_t(v; x)) + D_{\mathrm{KL}}(r_g(u; x) \| q_g(u|x)) \right] \end{array} \right) \quad \begin{array}{l}\text{Bias of AIDE} \\ \text{for } M_g = M_t = 1\end{array} \quad (3)
$$

Note that this expression involves KL divergences between the meta-inference sampling distributions ($r_g(u; x)$ and $r_t(v; x)$) and the posteriors in their respective generative inference models ($q_g(u|x)$ and $q_t(v|x)$). Therefore, the approximation error of meta-inference determines the bias of AIDE. When both meta-inference algorithms are exact ($r_g(u; x) = q_g(u|x)$ for all $u$ and $x$ and $r_t(v; x) = q_t(v|x)$ for all $v$ and $x$), AIDE is unbiased. As $M_g$ or $M_t$ are increased, the bias decreases (see Figure 2 and Figure 4 for examples). If the generative inference model for one of the algorithms does not use a trace (i.e. $\mathcal{U} = \{()\}$ or $\mathcal{V} = \{()\}$), then that algorithm does not contribute a KL divergence term to the bias of Equation (3). The analysis of AIDE is equivalent to that of Grosse et al. [12] when the target algorithm is AIS and $M_t = M_g = 1$ and the gold-standard inference algorithm is a rejection sampler.

## 4 Related Work

Diagnosing the convergence of approximate inference is a long-standing problem. Most existing work is either tailored to specific inference algorithms [13], designed to detect lack of exact convergence [1], or both. Estimators of the non-asymptotic approximation error of general approximate inference

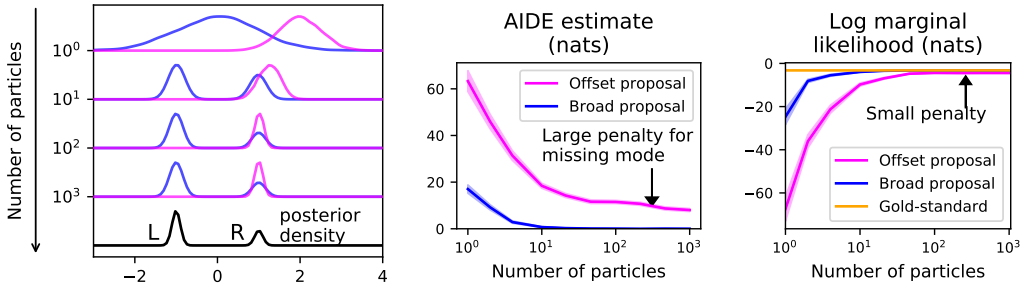

Figure 3: **AIDE detects when an inference algorithm misses a posterior mode**. Left: A bimodal posterior density, with kernel estimates of the output densities of importance sampling with resampling (SIR) using two proposals. The 'broad' proposal (blue) covers both modes, and the 'offset' proposal (pink) misses the 'L' mode. Middle: AIDE detects the missing mode in offset-proposal SIR. Right: Log marginal likelihood estimates suggest that the offset-proposal SIR is nearly converged.

algorithms have received less attention. Gorham and Mackey [14] propose an approach that applies to arbitrary sampling algorithms but relies on special properties of the posterior distribution such as log-concavity. Our approach does not rely on special properties of the posterior distribution.

Our work is most closely related to Bounding Divergences with REverse Annealing (BREAD, [12]) which also estimates upper bounds on the symmetric KL divergence between the output distribution of a sampling algorithm and the posterior distribution. AIDE differs from BREAD in two ways: First, whereas BREAD handles single-particle SMC samplers and annealed importance sampling (AIS), AIDE handles a substantially broader family of inference algorithms including SMC samplers with both resampling and rejuvenation steps, AIS, variational inference, and rejection samplers. Second, BREAD estimates divergences between the target algorithm's sampling distribution and the posterior distribution, but the exact posterior samples necessary for BREAD's theoretical properties are only readily available when the observations $y$ that define the inference problem are simulated from the generative model. Instead, AIDE estimates divergences against an exact or approximate gold-standard sampler on real (non-simulated) inference problems. Unlike BREAD, AIDE can be used to evaluate inference in both generative and undirected models.

AIDE estimates the error of sampling-based inference using a mathematical framework with roots in variational inference. Several recent works have treated sampling-based inference algorithms as variational approximations. The Monte Carlo Objective (MCO) formalism of Maddison et al. [15] is closely related to our formalism of generative inference models and meta-inference algorithms— indeed a generative inference model and a meta-inference algorithm with $Z = 1$ give an MCO defined by: $\mathcal{L}(y, p) = \mathbb{E}_{u,x\sim q(u,x)}[\log(p(x, y)/\xi(u, x))]$, where $y$ denotes observed data. In independent and concurrent work to our own, Naesseth et al. [16], Maddison et al. [15] and Le et al. [17] treat SMC as a variational approximation using constructions similar to ours. In earlier work, Salimans et al. [18] recognized that MCMC samplers can be treated as variational approximations. However, these works are concerned with optimization of variational objective functions instead of estimation of KL divergences, and do not involve generating a trace of a sampler from its output.

## 5   Experiments

### 5.1   Comparing the bias of AIDE for different types of inference algorithms

We used a Bayesian linear regression inference problem where exact posterior sampling is tractable to characterize the bias of AIDE when applied to three different types of target inference algorithms: sequential Monte Carlo (SMC), Metropolis-Hastings (MH), and variational inference. For the gold-standard algorithm we used a posterior sampler with a tractable output distribution $q_g(x)$, which does not introduce bias into AIDE, so that AIDE's bias could be completely attributed to the approximation error of meta-inference for each target algorithm. Figure 2 shows the results. The bias of AIDE is acceptable for SMC, and AIDE is unbiased for variational inference, but better meta-inference algorithms for MCMC are needed to make AIDE practical for estimating the accuracy of MH.

## 5.2 Evaluating approximate inference in a Hidden Markov model

We applied AIDE to measure the approximation error of SMC algorithms for posterior inference in a Hidden Markov model (HMM). Because exact posterior inference in this HMM is tractable via dynamic programming, we used this opportunity to compare AIDE estimates obtained using the exact posterior as the gold-standard with AIDE estimates obtained using a 'best-in-class' SMC algorithm as the gold-standard. Figure 4 shows the results, which indicate AIDE estimates using an approximate gold-standard algorithm can be nearly identical to AIDE estimates obtained with an exact posterior gold-standard.

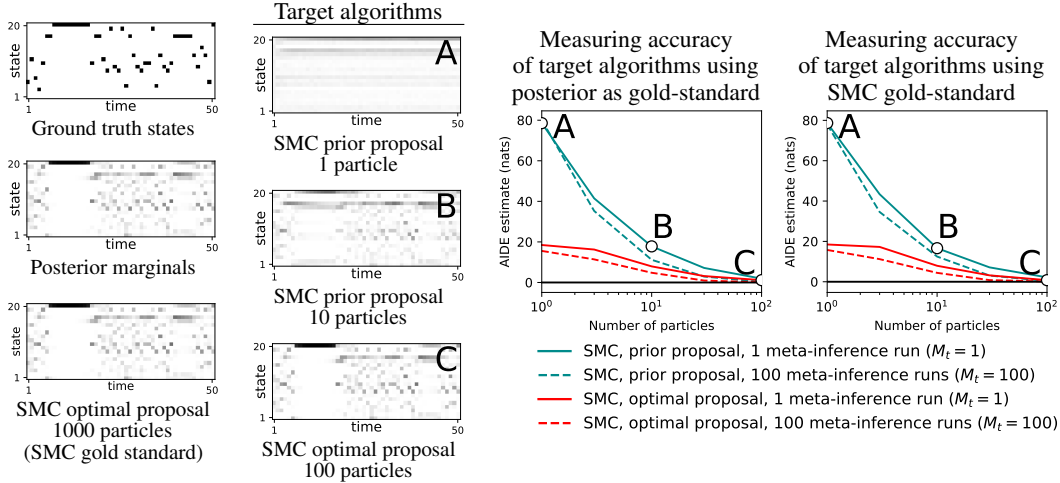

Figure 4: Comparing use of an exact posterior as the gold-standard and a 'best-in-class' approximate algorithm as the gold-standard, when measuring accuracy of target inference algorithms with AIDE. We consider inference in an HMM, so that exact posterior sampling is tractable using dynamic programming. Left: Ground truth latent states, posterior marginals, and marginals of the the output of a gold-standard and three target SMC algorithms (A,B,C) for a particular observation sequence. Right: AIDE estimates using the exact gold-standard and using the SMC gold-standard are nearly identical. The estimated divergence bounds decrease as the number of particles in the target sampler increases. The optimal proposal outperforms the prior proposal. Increasing $M_\mathrm{t}$ tightens the estimated divergence bounds. We used $M_\mathrm{g} = 1$.

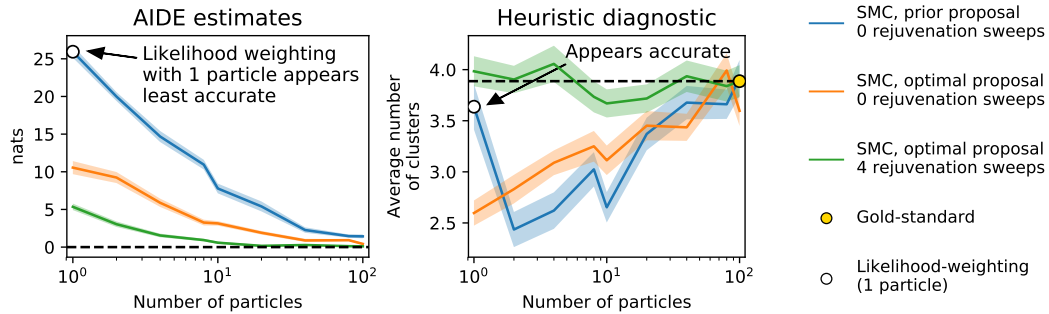

Figure 5: Contrasting AIDE against a heuristic convergence diagnostic for evaluating the accuracy of approximate inference in a Dirichlet process mixture model (DPMM). The heuristic compares the expected number of clusters under the target algorithm to the expectation under the gold-standard algorithm [19]. White circles identify single-particle likelihood-weighting, which samples from the prior. AIDE clearly indicates that single-particle likelihood-weighting is inaccurate, but the heuristic suggests it is accurate. Probe functions like the expected number of clusters can be error prone measures of convergence because they only track convergence along a specific projection of the distribution. In contrast, AIDE estimates a joint KL divergence. Shaded areas in both plots show the standard error. The amount of target inference computation used is the same for the two techniques, although AIDE performs a gold-standard meta-inference run for each target inference run.

### 5.3 Comparing AIDE to alternative inference evaluation techniques

A key feature of AIDE is that it applies to different types of inference algorithms. We compared AIDE to two existing techniques for evaluating the accuracy of inference algorithms that share this feature: (1) comparing log marginal likelihood (LML) estimates made by a target algorithm against LML estimates made by a gold-standard algorithm, and (2) comparing the expectation of a probe function under the approximating distribution to the same expectation under the gold-standard distribution [19]. Figure 3 shows a comparison of AIDE to LML, on a inference problem where the posterior is bimodal. Figure 5 shows a comparison of AIDE to a 'number of clusters' probe function in a Dirichlet process mixture model inference problem for a synthetic data set. We also used AIDE to evaluate the accuracy of several SMC algorithms for DPMM inference on a real data set of galaxy velocities [20] relative to an SMC gold-standard. This experiment is described in the supplement due to space constraints.

## 6 Discussion

AIDE makes it practical to estimate bounds on the error of a broad class of approximate inference algorithms including sequential Monte Carlo (SMC), annealed importance sampling (AIS), sampling importance resampling (SIR), and variational inference. AIDE's reliance on a gold-standard inference algorithm raises two questions that merit discussion:

*If we already had an acceptable gold-standard, why would we want to evaluate other inference algorithms?* Gold-standard algorithms such as very long MCMC runs, SMC runs with hundreds of thousands of particles, or AIS runs with a very fine annealing schedule, are often too slow to use in production. AIDE make it possible to use gold-standard algorithms during an offline design and evaluation phase to quantitatively answer questions like "how few particles or rejuvenation steps or samples can I get away with?" or "is my fast variational approximation good enough?". AIDE can thus help practitioners confidently apply Monte Carlo techniques in challenging, performance constrained applications, such as probabilistic robotics or web-scale machine learning. In future work we think it will be valuable to build probabilistic models of AIDE estimates, conditioned on features of the data set, to learn offline what problem instances are easy or hard for different inference algorithms. This may help practitioners bridge the gap between offline evaluation and production more rigorously.

*How do we ensure that the gold-standard is accurate enough for the comparison with it to be meaningful?* This is an intrinsically hard problem—we are not sure that near-exact posterior inference is really feasible, for most interesting classes of models. In practice, we think that gold-standard inference algorithms will be calibrated based on a mix of subjective assumptions and heuristic testing—much like models themselves are tested. For example, users could initially build confidence in a gold-standard algorithm by estimating the symmetric KL divergence from the posterior on simulated data sets (following the approach of Grosse et al. [12]), and then use AIDE with the trusted gold-standard for a focused evaluation of target algorithms on real data sets of interest. We do not think the subjectivity of the gold-standard assumption is a unique limitation of AIDE.

A limitation of AIDE is that its bias depends on the accuracy of meta-inference, i.e. inference over the auxiliary random variables used by an inference algorithm. We currently lack an accurate meta-inference algorithm for MCMC samplers that do not employ annealing, and therefore AIDE is not yet suitable for use as a general MCMC convergence diagnostic. Research on new meta-inference algorithms for MCMC and comparisons to standard convergence diagnostics [21, 22] are needed. Other areas for future work include understanding how the accuracy of meta-inference depends on parameters of an inference algorithm, and more generally what makes an inference algorithm amenable to efficient meta-inference.

Note that AIDE does not rely on asymptotic exactness of the inference algorithm being evaluated. An interesting area of future work is in using AIDE to study the non-asymptotic error of scalable but asymptotically biased sampling algorithms [23]. It also seems fruitful to connect AIDE to results from theoretical computer science, including the computability [24] and complexity [25–28] of probabilistic inference. It should be possible to study the computational tractability of approximate inference empirically using AIDE estimates, as well as theoretically using a careful treatment of the variance of these estimates. It also seems promising to use ideas from AIDE to develop Monte Carlo program analyses for samplers written in probabilistic programming languages.

**Acknowledgments**

This research was supported by DARPA (PPAML program, contract number FA8750-14-2-0004), IARPA (under research contract 2015-15061000003), the Office of Naval Research (under research contract N000141310333), the Army Research Office (under agreement number W911NF-13-1-0212), and gifts from Analog Devices and Google. This research was conducted with Government support under and awarded by DoD, Air Force Office of Scientific Research, National Defense Science and Engineering Graduate (NDSEG) Fellowship, 32 CFR 168a.

## Footnotes

[1]The trace data structure could in principle be obtained by writing the inference algorithm in a probabilistic programming language like Church [11], but the computational overhead would be high.

[2]AIDE also applies to approximate inference algorithms for undirected probabilistic models; the marginal likelihood estimate is replaced with the estimate of the partition function.

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
