[Supplementary Material]

# Supplementary material for 'AIDE: An algorithm for measuring the accuracy of probabilistic inference algorithms'

## A  Sequential Monte Carlo

An SMC sampler template based on [1] is reproduced in Algorithm 3. The algorithm evolves a set of $P$ particles to approximate a sequence of target distributions using a combination of proposal kernels, weighting, and resampling steps. The final target distribution in the sequence is typically the posterior $p(x|y)$. In our version, the algorithm resamples once from the final weighted particle approximation and returns this particle as its output sample $x$. Specifically, the algorithm uses a sequence of unnormalized target distributions $\tilde{p}_t$ defined on spaces $\mathcal{X}_t$ for $t = 1 \ldots T$, with $\mathcal{X}_T = \mathcal{X}$ (the original space of latent variables in the generative model) and $\tilde{p}_T(x) := p(x, y)$. The algorithm also makes use of an initialization kernel $k_1$ defined on $\mathcal{X}_1$, proposal kernels $k_t$ defined on $\mathcal{X}_t$ and indexed by $\mathcal{X}_{t-1}$ for $t = 2 \ldots T$, and backward kernels $\ell_t$ defined on $\mathcal{X}_{t-1}$ and indexed by $\mathcal{X}_t$ for $t = 2 \ldots T$. For simplicity of analysis we assume that resampling occurs at every step in the sequence. The weight functions used in the algorithm are:

$$w_1(x_1^i) := \frac{\tilde{p}_1(x_1^i)}{k_1(x_1^i)} \qquad w_t(x_{t-1}^j, x_t^i) := \frac{\tilde{p}_t(x_t^i)\ell_t\left(x_{t-1}^j; x_t^i\right)}{\tilde{p}_{t-1}\left(x_{t-1}^j\right) k_t\left(x_t^i; x_{t-1}^j\right)} \tag{1}$$

Note that Algorithm 3 does not sample from the backward kernels, which serve to define the extended target distributions $\tilde{p}_t(x_t) \prod_{s=2}^t \ell_s(x_{s-1}; x_s)$ that justify the SMC sampler as a sequential importance sampler [1]. When $P = 1$, $\mathcal{X}_t = \mathcal{X}$ for all $t$, $k_t(x_t; x_{t-1})$ is a detailed balance transition operator for $p_{t-1}$, and $\ell_t = k_t$, the algorithm reduces to AIS.[1] The particle filter without rejuvenation [3] is also a special case of Algorithm 3. A variety of other SMC variants can also be seen to be special cases of this formulation [1]. The SMC marginal likelihood estimate $\widehat{p(y)}$ is computed from the weights $w_t^i$ generated during the SMC algorithm according to:

$$\widehat{p(y)} = \prod_{t=1}^{T} \frac{1}{P} \sum_{i=1}^{P} w_t^i \tag{2}$$

Note that an SMC marginal likelihood estimate can also be computed from the weights $w_t^i$ generated during the generalized conditional SMC algorithm. Note that $\widehat{p(y)}$ a function of the SMC trace $u$. To relate $\xi(u, x)$ to $\widehat{p(y)}$, we write the joint distribution of the generative inference model for SMC:

$$q(u, x) := \left[\prod_{i=1}^{P} k_1(x_1^i)\right] \left[\prod_{t=2}^{T} \prod_{i=1}^{P} \frac{w_{t-1}^{a_{t-1}^i}}{\sum_{j=1}^{P} w_{t-1}^j} k_t(x_t^i; x_{t-1}^{a_{t-1}^i})\right] \left[\frac{w_T^{I_T}}{\sum_{j=1}^{P} w_T^j}\right] \delta(x, x_T^{I_T}) \tag{3}$$

**Algorithm 3** Sequential Monte Carlo

> **for** $i \leftarrow 1 \dots P$ **do**
>      $x_1^i \sim k_1(\cdot)$          ▷ Initialize particle $i$
>      $w_1^i \leftarrow w_1(x_1^i)$          ▷ Initial weight for particle $i$
> **for** $t \leftarrow 2 \dots T$ **do**
>      $W_{t-1}^{1:P} \leftarrow w_{t-1}^{1:P}/(\sum_{i=1}^P w_{t-1}^i)$          ▷ Normalize weights $w_{t-1}^{1:P} = (w_{t-1}^1, \dots, w_{t-1}^P)$
>      **for** $i \leftarrow 1 \dots P$ **do**
>          $a_{t-1}^i \sim \text{Categorical}(W_{t-1}^{1:P})$          ▷ Sample index of parent for particle $i$
>          $x_t^i \sim k_t\left(\cdot; x_{t-1}^{a_{t-1}^i}\right)$          ▷ Sample value for new particle $i$
>          $w_t^i \leftarrow w_t(x_{t-1}^{a_{t-1}^i}, x_t^i)$          ▷ Compute weight for particle $i$
> $I_T \sim \text{Categorical}(W_T^{1:P})$          ▷ Sample particle index for output sample
> $x \leftarrow x_T^{I_T}$
> **return** $x$          ▷ Return the output sample

The the canonical meta-inference sampler (Algorithm 1) for SMC takes as input a latent sample $x$ and returns a trace $u = (\mathbf{x}, \mathbf{a}, I_T)$ of Algorithm 3, containing all particles at all time steps $\mathbf{x}$, all parent indices $\mathbf{a}$, and the final output particle index $I_T$. The distribution on outputs of the meta-inference sampler is given by:

$$r(u; x) := \delta(x_T^{I_T}, x) \frac{1}{P^T} \left[ \prod_{t=2}^T \ell_t(x_{t-1}^{I_{t-1}}; x_t^{I_t}) \right] \left[ \prod_{\substack{i=1 \\ i \neq I_1}}^P k_1(x_1^i) \right] \left[ \prod_{t=2}^T \prod_{\substack{i=1 \\ i \neq I_t}}^P \frac{w_{t-1}^{a_{t-1}^i}}{\sum_{j=1}^P w_{t-1}^j} k_t(x_t^i; x_{t-1}^{a_{t-1}^i}) \right] \quad (4)$$

Taking the ratio $q(u, x)/r(u; x)$ and simplifying gives $p(x, y)/\widehat{p(y)}$. Therefore, the quantity $\xi(u, x) = q(u, x)/r(u; x)$ can be computed from an SMC trace $u$, in terms of the SMC marginal likelihood estimate and the unnormalized posterior probability $p(x, y)$.

# B   AIDE specialized for evaluating variational inference using AIS

To make AIDE more concrete for the reader, we provide the AIDE algorithm when specialized to measure the symmetrized KL divergence between a variational approximation $q_\theta(x)$ and an annealed importance sampler (AIS). For variational inference, there is no meta-inference sampler necessary because we can evaluate the variational approximating distribution, as discussed in the main text. The meta-inference sampler for AIS consists of running the AIS chain in reverse, starting from a latent sample. The trace $u$ that is generated is the vector of intermediate states $\mathbf{x} = (x_1, \dots, x_T)$ in the chain. The AIS marginal likelihood estimate $\widehat{p(y)}$ can be computed from a trace $u$ of the AIS algorithm in terms of the weights (which are themselves deterministic functions of the trace):

$$\widehat{p(y)} = \frac{\tilde{p}_1(x_1)}{k_1(x_1)} \prod_{t=2}^T \frac{\tilde{p}_t(x_t)}{\tilde{p}_{t-1}(x_t)} \quad (5)$$

Note that $\widehat{p(y)}$ can be computed from a trace $u$ that is generated either by a 'forward' run of AIS, or a reverse run of AIS. Algorithm 4 gives a concrete instantiation of AIDE (Algorithm 2) simplified for the case when the gold-standard algorithm is an AIS sampler, and the target algorithm being

evaluated is a variational approximation. We further simplify the algorithm by fixing $M_g = 1$, where AIS is the gold-standard. The AIS algorithm must support two primitives: AIS.FORWARD(), which runs AIS forward and returns the resulting output sample $x$ and the resulting marginal likelihood estimate $\widehat{p(y)}$, and AIS.REVERSE($x$), which takes as input a sample $x$ and runs the same AIS chain in reverse order, returning the resulting marginal likelihood estimate $\widehat{p(y)}$.

---

**Algorithm 4** AIDE specialized for measuring divergence between variational inference and AIS

---

**Require:**  AIS algorithm                            AIS.FORWARD() and AIS.REVERSE($x$)
                 Trained variational approximation    $q_\theta(x)$
                 Number of AIS forward samples       $N_g$
                 Number of variational samples       $N_t$

  **for** $n \leftarrow 1 \dots N_g$ **do**
       ▷ Run AIS forward, record the marginal likelihood estimate $\widehat{p(y)}_n$ and the final state in chain $x_n$
       $\left( \widehat{p(y)}_n, x_n \right) \sim$ AIS.FORWARD()

  **for** $n \leftarrow 1 \dots N_t$ **do**
       ▷ Generate sample $x'_n$ from the variational approximation
       $x'_n \sim q_\theta(x)$
       ▷ Run AIS in reverse, starting from $x'_n$, and record resulting marginal likelihood estimate $\widehat{p(y)}'_n$
       $\widehat{p(y)}'_n \sim$ AIS.REVERSE($x'_n$)

  ▷ Compute AIDE estimate
  $\hat{D} \leftarrow \dfrac{1}{N_g} \sum_{n=1}^{N_g} \log \left( \dfrac{p(x_n, y)}{q_\theta(x_n)\widehat{p(y)}_n} \right) - \dfrac{1}{N_t} \sum_{n=1}^{N_t} \log \left( \dfrac{p(x'_n, y)}{q_\theta(x'_n)\widehat{p(y)}'_n} \right)$
  **return** $\hat{D}$

---

# C   Proofs

**Theorem 1.** *The estimate $\hat{D}$ produced by AIDE is an upper bound on the symmetrized KL divergence in expectation, and the expectation is nonincreasing in AIDE parameters $M_g$ and $M_t$.*

*Proof.* We consider the general case of two inference algorithms $a$ and $b$ with generative inference models $(\mathcal{U}, \mathcal{X}, q_a)$ and $(\mathcal{V}, \mathcal{X}, q_b)$, and meta-inference algorithms $(r_a, \xi_a)$ and $(r_b, \xi_b)$ with normalizing constants $Z_a$ and $Z_b$ respectively. For example $a$ may be 'target' inference algorithm and $b$ may be the 'gold standard' inference algorithm. Note that the analysis of AIDE is symmetric in $a$ and $b$. First, we define the following quantity relating $a$ and $b$:

$$\mathcal{L}_{ab} := \mathbb{E}_{x \sim q_a(x)} \left[ \log \frac{Z_b q_b(x)}{Z_a q_a(x)} \right] = \log \frac{Z_b}{Z_a} - D_{\mathrm{KL}}(q_a(x) \parallel q_b(x)) \tag{6}$$

When $Z_a = 1$ and when $b$ is a rejection sampler for the posterior $p(x|y)$, we have that $Z_b = p(x, y)$, and $\mathcal{L}_{ab}$ is the 'ELBO' of inference algorithm $a$ with respect to the posterior. We also define the quantity:

$$\mathcal{U}_{ab} := -\mathcal{L}_{ba} = \log \frac{Z_b}{Z_a} + D_{\mathrm{KL}}(q_b(x) \parallel q_a(x)) \tag{7}$$

Note that $\mathcal{U}_{ab} - \mathcal{L}_{ab} = \mathcal{U}_{ba} - \mathcal{L}_{ba}$ is the symmetrized KL divergence between $q_a(x)$ and $q_b(x)$. The AIDE estimate can be understood as a difference of an estimate of $\mathcal{U}_{ab}$ and an estimate of $\mathcal{L}_{ab}$.

Specifically, we define the following estimator of $\mathcal{L}_{ab}$:

$$\hat{\mathcal{L}}_{ab}^{N_a,M_a,M_b} := \frac{1}{N_a} \sum_{n=1}^{N_a} \log \left( \frac{\frac{1}{M_b} \sum_{k=1}^{M_b} \xi_b(v_{n,k}, x_n)}{\frac{1}{M_a} \sum_{k=1}^{M_a} \xi_a(u_{n,k}, x_n)} \right) \tag{8}$$

$$= \frac{1}{N_a} \sum_{n=1}^{N_a} \log \left( \frac{\frac{1}{M_b} \sum_{k=1}^{M_b} Z_b \frac{q_b(v_{n,k},x_n)}{r_b(v_{n,k};x_n)}}{\frac{1}{M_a} \sum_{k=1}^{M_a} Z_a \frac{q_a(u_{n,k},x_n)}{r_a(u_{n,k};x_n)}} \right) \tag{9}$$

where:
$$x_n \sim q_a(x) \text{ for } n = 1 \ldots N_a$$
$$u_{n,1}|x_n \sim q_a(u|x) \text{ for } n = 1 \ldots N_a$$
$$u_{n,k}|x_n \sim r_a(u;x) \text{ for } n = 1 \ldots N_a \text{ and } k = 2 \ldots M_a$$
$$v_{n,k}|x_n \sim r_b(v;x) \text{ for } n = 1 \ldots N_a \text{ and } k = 1 \ldots M_b$$

We now analyze the expectation $\mathbb{E}[\hat{\mathcal{L}}_{ab}^{1,M_a,M_b}]$ and how it depends on $M_a$ and $M_b$. We use the notation $u_{i:j} = (u_i, \ldots, u_j)$. First, note that:

$$\mathbb{E}[\hat{\mathcal{L}}_{ab}^{1,M_a,M_b}] = \log \frac{Z_b}{Z_a} + \mathbb{E}_{x\sim q_a(x)} \left[ \log \frac{q_b(x)}{q_a(x)} \right]$$

$$+ \mathbb{E}_{\substack{x\sim q_a(x) \\ v_{1:M_b}|x \overset{iid}{\sim} r_b(v;x)}} \left[ \log \frac{1}{M_b} \sum_{k=1}^{M_b} \frac{q_b(v_k|x)}{r_b(v_k;x)} \right]$$

$$- \mathbb{E}_{\substack{x\sim q_a(x) \\ u_1|x\sim q_a(u|x) \\ u_{2:M_a}|x \overset{iid}{\sim} r_a(u;x)}} \left[ \log \frac{1}{M_a} \sum_{k=1}^{M_a} \frac{q_a(u_k|x)}{r_a(u_k;x)} \right] \tag{10}$$

$$= \mathcal{L}_{ab} + \mathbb{E}_{\substack{x\sim q_a(x) \\ v_{1:M_b}|x \overset{iid}{\sim} r_b(v;x)}} \left[ \log \frac{1}{M_b} \sum_{k=1}^{M_b} \frac{q_b(v_k|x)}{r_b(v_k;x)} \right]$$

$$- \mathbb{E}_{\substack{x\sim q_a(x) \\ u_1\sim q_a(u|x) \\ u_{2:M_a}|x \overset{iid}{\sim} r_a(u;x)}} \left[ \log \frac{1}{M_a} \sum_{k=1}^{M_a} \frac{q_a(u_k|x)}{r_a(u_k;x)} \right] \tag{11}$$

We define the following families of distributions on $v_{1:M_b}$, indexed by $x$:

$$\eta_b^{M_b}(v_{1:M_b}; x) = \frac{1}{M_b} \sum_{k=1}^{M_b} q_b(v_k|x) \prod_{\substack{\ell=1 \\ \ell \neq k}}^{M_b} r_b(v_\ell; x) \tag{12}$$

$$\lambda_b^{M_b}(v_{1:M_b}; x) = \prod_{k=1}^{M_b} r_b(v_k; x) \tag{13}$$

and similarly for $u_{1:M_a}$:

$$\eta_a^{M_a}(v_{1:M_a}; x) = \frac{1}{M_a} \sum_{k=1}^{M_a} q_a(u_k|x) \prod_{\substack{\ell=1 \\ \ell \neq k}}^{M_a} r_a(u_\ell; x) \tag{14}$$

$$\lambda_a^{M_a}(u_{1:M_a}; x) = \prod_{k=1}^{M_a} r_a(u_k; x) \tag{15}$$

Taking the first expectation in Equation (11):

$$\mathbb{E}_{\substack{x \sim q_a(x) \\ v_{1:M_b} \overset{iid}{\sim} r_b(v;x)}} \left[ \log \frac{1}{M_b} \sum_{k=1}^{M_b} \frac{q_b(v_k|x)}{r_b(v_k;x)} \right] \tag{16}$$

$$= \mathbb{E}_{\substack{x \sim q_a(x) \\ v_{1:M_b} \overset{iid}{\sim} r_b(v;x)}} \left[ \log \frac{\frac{1}{M_b} \sum_{k=1}^{M_b} q_b(v_k|x) \prod_{\substack{\ell=1 \\ \ell \neq k}}^{M_b} r_b(v_\ell;x)}{\prod_{k=1}^{M_b} r_b(v_k;x)} \right] \tag{17}$$

$$= \mathbb{E}_{\substack{x \sim q_a(x) \\ v_{1:M_b} \sim \lambda_b^{M_b}(v_{1:M_b};x)}} \left[ \log \frac{\eta_b^{M_b}(v_{1:M_b};x)}{\lambda_b^{M_b}(v_{1:M_b};x)} \right] \tag{18}$$

$$= -\mathbb{E}_{x \sim q_a(x)} \left[ D_{\mathrm{KL}}(\lambda_b^{M_b}(v_{1:M_b};x) \,\|\, \eta_b^{M_b}(v_{1:M_b};x)) \right] \tag{19}$$

Taking the second expectation in Equation (11):

$$\mathbb{E}_{\substack{x \sim q_a(x) \\ u_1 \sim q_a(u|x) \\ u_{2:M_a} \overset{iid}{\sim} r_a(u;x)}} \left[ \log \frac{1}{M_a} \sum_{k=1}^{M_a} \frac{q_a(u_k|x)}{r_a(u_k;x)} \right] \tag{20}$$

$$= \mathbb{E}_{\substack{x \sim q_a(x) \\ u_1 \sim q_a(u|x) \\ u_{2:M_a} \overset{iid}{\sim} r_a(u;x)}} \left[ \log \frac{\frac{1}{M_a} \sum_{k=1}^{M_a} q_a(u_k|x) \prod_{\substack{\ell=1 \\ \ell \neq k}}^{M_a} r_a(u_\ell;x)}{\prod_{k=1}^{M_a} r_a(u_k;x)} \right] \tag{21}$$

$$= \mathbb{E}_{\substack{x \sim q_a(x) \\ u_1 \sim q_a(u|x) \\ u_{2:M_a} \overset{iid}{\sim} r_a(u;x)}} \left[ \log \frac{\eta_a^{M_a}(u_{1:M_a};x)}{\lambda_a^{M_a}(u_{1:M_a};x)} \right] \tag{22}$$

$$= \mathbb{E}_{\substack{x \sim q_a(x) \\ u_{1:M_a} \sim \eta_a^{M_a}(u_{1:M_a};x)}} \left[ \log \frac{\eta_a^{M_a}(u_{1:M_a};x)}{\lambda_a^{M_a}(u_{1:M_a};x)} \right] \tag{23}$$

$$= \mathbb{E}_{x \sim q_a(x)} \left[ D_{\mathrm{KL}}(\eta_a^{M_a}(u_{1:M_a};x) \,\|\, \lambda_a^{M_a}(u_{1:M_a};x)) \right] \tag{24}$$

where to obtain Equation (23) we used the fact that $\log(\eta_a^{M_a}(u_{1:M_a};x)/\lambda_a^{M_a}(u_{1:M_a};x))$ is invariant to permutation of its arguments $u_{1:M_a}$. Substituting the expression given by Equation (19) and the expression given by Equation (24) into Equation (11), we have:

$$\mathbb{E}[\hat{\mathcal{L}}_{ab}^{1,M_a,M_b}] = \mathcal{L}_{ab} - \mathbb{E}_{x \sim q_a(x)} \left[ D_{\mathrm{KL}}(\lambda_b^{M_b}(v_{1:M_b};x) \,\|\, \eta_b^{M_b}(v_{1:M_b};x)) \right] \tag{25}$$

$$- \mathbb{E}_{x \sim q_a(x)} \left[ D_{\mathrm{KL}}(\eta_a^{M_a}(u_{1:M_a};x) \,\|\, \lambda_a^{M_a}(u_{1:M_a};x)) \right] \tag{26}$$

From non-negativity of KL divergence:

$$\mathbb{E}[\hat{\mathcal{L}}_{ab}^{1,M_a,M_b}] \leq \mathcal{L}_{ab} \tag{27}$$

Next, we show that $\mathbb{E}[\hat{\mathcal{L}}_{ab}^{1,M_a,M_b}]$ is nondecreasing in both $M_a$ and $M_b$. First, we show this for $M_a$. We introduce the notation $u_{1:k-1:k+1:M_a} := (u_1, \ldots, u_{k-1}, u_{k+1}, \ldots, M_a)$ to denote the subvector of length $M_a - 1$ obtained by removing element $k$ from vector $u_{1:M_a}$, where $u_{1:0:2:M_a} := u_{2:M_a}$ and $u_{1:M_a-1:M_a+1:M_a} := u_{1:M_a-1}$. Note that:

$$\eta_a^{M_a}(u_{1:M_a};x) = \frac{1}{M_a} \sum_{k=1}^{M_a} \eta_a^{M_a-1}(u_{1:k-1:k+1:M_a};x) r_a(u_k;x) \tag{28}$$

By convexity of KL divergence, we have:

$$D_{\mathrm{KL}}(\eta_a^{M_a}(u_{1:M_a};x) \parallel \lambda_a^{M_a}(u_{1:M_a};x)) \tag{29}$$

$$\leq \frac{1}{M_a} \sum_{k=1}^{M_a} D_{\mathrm{KL}}(\eta_a^{M_a-1}(u_{1:k-1:k+1:M_a};x)r_a(u_k;x) \parallel \lambda_a^{M_a}(u_{1:M_a};x)) \tag{30}$$

$$= \frac{1}{M_a} \sum_{k=1}^{M_a} D_{\mathrm{KL}}(\eta_a^{M_a-1}(u_{1:M_a-1};x) \parallel \lambda_a^{M_a-1}(u_{1:M_a-1};x)) \tag{31}$$

$$= D_{\mathrm{KL}}(\eta_a^{M_a-1}(u_{1:M_a-1};x) \parallel \lambda_a^{M_a-1}(u_{1:M_a-1};x)) \tag{32}$$

A similar argument can be used to show that:

$$D_{\mathrm{KL}}(\lambda_b^{M_b}(v_{1:M_b};x) \parallel \eta_b^{M_b}(v_{1:M_b};x)) \tag{33}$$

$$\leq D_{\mathrm{KL}}(\lambda_b^{M_b-1}(v_{1:M_b-1};x) \parallel \eta_b^{M_b-1}(v_{1:M_b-1};x)) \tag{34}$$

Applying these inequalities to Equation (26), we have:

$$\mathcal{L}_{ab} \geq \mathbb{E}[\hat{\mathcal{L}}_{ab}^{1,M_a,M_b}] \geq \mathbb{E}[\hat{\mathcal{L}}_{ab}^{1,M_a-1,M_b}] \tag{35}$$

$$\mathcal{L}_{ab} \geq \mathbb{E}[\hat{\mathcal{L}}_{ab}^{1,M_a,M_b}] \geq \mathbb{E}[\hat{\mathcal{L}}_{ab}^{1,M_a,M_b-1}] \tag{36}$$

To conclude the proof we apply these inequalities to the expectation of the AIDE estimate:

$$\hat{D}^{N_a,N_b,M_a,M_b} = -\hat{\mathcal{L}}_{ab}^{N_a,M_a,M_b} - \hat{\mathcal{L}}_{ba}^{N_b,M_b,M_a} \tag{37}$$

$$\mathbb{E}[\hat{D}^{N_a,N_b,M_a,M_b}] = \mathbb{E}[-\hat{\mathcal{L}}_{ab}^{1,M_a,M_b}] + \mathbb{E}[-\hat{\mathcal{L}}_{ba}^{1,M_b,M_a}] \tag{38}$$

$$\geq -\mathcal{L}_{ab} - \mathcal{L}_{ba} \tag{39}$$

$$= D_{\mathrm{KL}}(q_a(x) \parallel q_b(x)) + D_{\mathrm{KL}}(q_b(x) \parallel q_a(x)) \tag{40}$$

$$\mathbb{E}[\hat{D}^{N_a,N_b,M_a,M_b}] \leq \mathbb{E}[\hat{D}^{N_a,N_b,M_a-1,M_b}] \tag{41}$$

$$\mathbb{E}[\hat{D}^{N_a,N_b,M_a,M_b}] \leq \mathbb{E}[\hat{D}^{N_a,N_b,M_a,M_b-1}] \tag{42}$$

$$\square$$

# D  Bias of AIDE for AIS and MH

When the generic SMC algorithm (Algorithm 3) is used with a single particle ($P = 1$), the algorithm becomes a Markov chain that samples from transition kernels $k_t$, and the canonical SMC meta-inference algorithm also becomes a Markov chain that samples from transition kernels $\ell_t$ in reverse order. For this analysis we assume that $k_t = \ell_t$ and that $k_t$ satisfies detailed balance with respect to intermediate distribution $p_{t-1}$ for $t = 2 \ldots T$. Then, the incremental weight simplifies to:

$$w_t(x_{t-1}, x_t) = \frac{\tilde{p}_t(x_t)k_t(x_{t-1};x_t)}{\tilde{p}_{t-1}(x_{t-1})k_t(x_t;x_{t-1})} \tag{43}$$

$$= \frac{\tilde{p}_t(x_t)}{\tilde{p}_{t-1}(x_{t-1})} \frac{\tilde{p}_{t-1}(x_{t-1})}{\tilde{p}_{t-1}(x_t)} \tag{44}$$

$$= \frac{\tilde{p}_t(x_t)}{\tilde{p}_{t-1}(x_t)} \tag{45}$$

Under the limiting assumption that $k_t(x_t; x_{t-1}) = p_{t-1}(x_t)$, the approximation error of the canonical meta-inference algorithm becomes:

$$D_{\mathrm{KL}}(q(u|x) \parallel r(u;x)) = \sum_{t=2}^{T} D_{\mathrm{KL}}(p_{t-1}(x) \parallel p_t(x)) \tag{46}$$

where $p_0$ is defined to be the initialization distribution $k_1$, and where $p_T(x) = p(x|y)$. A similar result can be obtained for the other direction of divergence. If the intermediate distributions are sufficiently fine-grained, then empirically this divergence converges to zero (as demonstrated in e.g. [4]). However, in standard Markov chain Monte Carlo practice, without annealing, the intermediate distributions are $p_t = p_T$ for all $t > 0$. In this case, the approximation error of meta-inference is the divergence between the initializing distribution and the posterior, which is generally large. Better meta-inference algorithms that do not rely on the AIS assumption that the chain is near equilibrium at all times are needed in order for AIDE to be a practical tool for measuring the accuracy of standard, non-annealed Markov chain Monte Carlo.

# E    Evaluating SMC inference in DPMM for galaxy velocity data

We obtained a data set of galaxy velocities based on redshift [5], and randomly subsampled down to forty of the galaxies for analysis. A histogram of the data set is shown in Figure 1(a). We consider the task of inference in a collapsed normal-inverse-gamma DPMM. We used SMC with 100 particles, optimal (Gibbs) proposal for cluster assignments, and Metropolis-Hastings rejuvenation kernels over hyperparameters and Gibbs kernels over cluster assignments as the gold-standard inference algorithm. Using this gold-standard, we evaluated the accuracy of SMC inference with the prior proposal, and without rejuvenation kernels using AIDE and using an alternative diagnostic based on comparing the average number of clusters in the sampling distribution relative to the average number under the gold-standard sampling distribution. Results are shown in Figure 1(b) and Figure 1(c).

Figure 1: (a) shows a histogram of velocities of galaxies from [5]. We model this data set using a Dirichlet process mixture, and evaluate the accuracy of SMC inference algorithms relative to a gold-standard, using AIDE and using a heuristic diagnostic based on measuring the average number of clusters in the approximating distribution and the gold-standard distribution. (b) shows results of AIDE. (c) shows result of the heuristic diagnostic. Both techniques indicate that rejuvenation kernels are important for fast convergence. Unlike the heuristic diagnostic, AIDE does not require custom design of a probe function for each model. We envision AIDE being used in concert with heuristic diagnostics like (c). In our experience, AIDE provides more conservative quantification of accuracy than heuristic diagnostics. The experiment was performed on a subsampled set of 40 data points from the data set in (a).

## Footnotes

[1]More generally, the proposal kernel $k_t$ needs to have stationary distribution $p_{t-1}$. The backward kernel $\ell_t$ is the 'reversal' of $k_t$ as defined in [2]. When the proposal kernel satisfies detailed balance, it is its own reversal, and therefore sampling from the backward kernel is identical to sampling from the forward kernel.