[Reviews · NeurIPS 2017]

Reviewer 1



[after feedback] Thank you for the clarifications. I've updated my score. I do still have some concerns regarding applicability, since a gold standard "ground truth" of inference is required. I do appreciate the situation described in the feedback, where one is trying to decide on some approximate algorithm to "deploy" in the wild, is actually fairly common. That is, the sort of setting where very slow MCMC can be run for a long time on training data, but on new data, where there is e.g. a real-time requirement, a faster approximate inference algorithm will be used instead. [original review] This paper introduces a new method for benchmarking the performance of different approximate inference algorithms. The approach is based on constructing an estimator of the “symmetric” KL divergence (i.e., the sum of the forward and reverse KL) between an approximation to the target distribution and a representation of the “true” exact target distribution. The overall approach considered is interesting, and for the most part clearly presented. I would agree that there is lots of potential in approaches which directly consider the auxiliary random variables which occur within an approximate inference algorithm. The meta-inference performed here (as described in definition 3.2) relates estimation of the probability density assigned to a particular algorithm output to a marginal likelihood computation, marginalizing out the latent variables in the algorithm; SMC and AIS are then natural choices for meta-inference algorithms. The primary drawback of this paper is that the proposed divergence estimate D-hat provided by AIDE does not seem to lend itself to use as a practical diagnostic tool. This symmetric KL does not allow us to (say) decide whether one particular inference algorithm is “better” than another, it only allows us to characterize the discrepancy between two algorithms. In particular, if we lack “gold standard” inference for a particular problem, then it is not clear what we can infer from the D-hat — plots such as those in figure 3 can only be produced contingent on knowing which (of possibly several) inference algorithms is unambiguously the “best”. Do we really need a diagnostic such as this for situations where “gold standard” inference is computationally feasible to perform? I would appreciate it greatly if the authors could clarify how they expect to see AIDE applied. Figure 5 clearly demonstrates that the AIDE estimate is a strong diagnostic for the DPMM, but unless I am mistaken this cannot be computed online — only relative to separate gold-standard executions. The primary use case for this then seems to be for providing a tool for designers of approximate inference algorithms to accurately benchmark their own algorithm performance (which is certainly very useful! but, I don’t see e.g. how figure 5 fits in that context). I would additionally be concerned about the relative run-time cost of inference and meta-inference. For example, in the example shown in figure 4, we take SMC with 1000 particles and the optimal proposal as the gold standard — how does the computational cost of the gold standard compare to that of e.g. the dashed red line for SMC with optimal proposal and 100 meta-inference runs, at 10 or 100 particles? Minor questions: • Why are some of the inner for-loops in algorithm 2 beginning from m = 2, …, M, not from m = 1?